# Identification of KIFC3 as a Colorectal Cancer Biomarker and Its Regulatory Mechanism in the Immune Microenvironment Based on Integrated Analysis of Multi-Omics Databases

**DOI:** 10.3390/biomedicines13040859

**Published:** 2025-04-02

**Authors:** Fen Wang, Xinxin Zeng, Jielun Wen, Kexin Xian, Feng Jin, Sunfang Jiang, Liyue Sun

**Affiliations:** 1Department of Medical, Peking University Shenzhen Hospital, Shenzhen 518000, China; fina_wang@163.com (F.W.); jinfeng201@163.com (F.J.); 2Second Department of Oncology, Guangdong Second Provincial General Hospital, Guangzhou 518025, China; zxx97@stu2020.jnu.edu.cn (X.Z.); xkxmzh@126.com (K.X.); 3School of Medicine, Jinan University, Guangzhou 510632, China; wenjl2003@stu2021.jnu.edu.cn; 4Department of Health Management Centre, Zhongshan Hospital, Fudan University, Shanghai 200032, China; 5Department of General Practice, Zhongshan Hospital, Fudan University, Shanghai 200032, China

**Keywords:** colorectal cancer, kinesin family member, ScRNA-seq, KIFC3

## Abstract

**Background:** Studies suggest that kinesin family (KIF) members can promote the occurrence of colorectal cancer (CRC). However, the mechanism of action has not yet been elucidated. The aim of this study was to identify CRC biomarkers associated with KIF members and to investigate their biological mechanisms in the treatment of colorectal cancer by analyzing multi-omics data. **Methods:** CRC-related datasets and KIF member-related genes (KIFRGs) were used. First, differentially expressed genes (DEGs) and differentially expressed methylation genes (DEMGs) in the TCGA-CRC were identified separately using different expression analyses (CRC vs. control). The intersecting genes were selected by overlapping the DEGs, DEMGs, and KIFRGs. Candidate genes were identified using survival analysis (*p* < 0.05). Subsequently, based on the candidate genes, biomarkers were selected by gene expression validation and survival analysis. Subsequently, functional enrichment, immune cell infiltration, and drug sensitivity analyses were performed. Single-cell analysis was utilized to perform cell annotation, and then function enrichment and pseudo-temporal analyses were performed. **Results:** The 12 intersecting genes were identified by overlapping 12,479 DEGs, 11,319 DEMGs, and 43 KIFRGs. The survival analysis showed that Kinesin Family Member C2 (KIFC2) and Kinesin Family Member C3 (KIFC3) had significant differences in survival (*p* < 0.05). Moreover, KIFC3 passed the gene expression validation and survival analysis validation (*p* < 0.05); thus, KIFC3 was deemed a biomarker. Subsequently, the pathways involved in KIFC3 were detected, such as the Ecm receptor intersection and chemokine signaling pathway. In addition, we found that KIFC3 was significantly positively correlated with natural killer (NK) cells (r = 0.455, *p* < 0.05) and NK T cells (r = 0.411, *p* < 0.05). Moreover, in the drug sensitivity of the CRC, the potential therapeutic benefits of AZD.2281, nilotinib, PD.173074, and shikonin were detected. Furthermore, using single-cell analysis, 16 cell clusters were annotated, and epithelial cells and M2-like macrophages were enriched in “rheumatoid arthritis”. Additionally, we observed that most M1-like macrophages were present in the early stages of differentiation, whereas M2-like macrophages were predominant in the later stages of differentiation. **Conclusions:** This study identifies KIFC3 as a CRC biomarker through multi-omics analysis, highlighting its unique expression, survival association, immune correlations, and drug sensitivity for potential diagnostic and therapeutic applications.

## 1. Introduction

Colorectal cancer (CRC) is one of the most common malignant tumors of the digestive system. Its incidence has increased significantly worldwide, and this increase is closely related to lifestyle, dietary patterns, and environmental factors [1]. Despite continual advancements in medical technology, the primary diagnostic method for CRC remains invasive tissue biopsy via colonoscopy, which imposes substantial patient discomfort, prolonged recovery periods, and requires skilled medical personnel [2]. Moreover, conventional CRC diagnostic markers such as carbohydrate antigen 19-9 (CA19-9) exhibit limited sensitivity for early detection [3]. Consequently, there is an urgent need for a more precise and minimally invasive diagnostic approach to enhance early detection rates and treatment efficacy for CRC. This study aimed to identify and analyze CRC biomarkers of CRC, providing a reference for the treatment and prognosis management of CRC.

Members of the kinesin family (KIF) of proteins, a pivotal intracellular transport molecular motor, plays crucial roles in mitosis, material conveyance, and other physiological processes [4]. Beyond their involvement in normal cell division, they participate in malignant processes such as tumor cell proliferation, migration, and invasion [5]. For instance, elevated expression of KIF26B has been observed in CRC, and its inhibition can suppress CRC cell proliferation [6]. KIF11 is upregulated in CRC, and its silencing not only inhibits tumor growth but also increases CRC cell sensitivity to oxaliplatin via the p53/GSK3β signaling pathway [7]. Additionally, KIFC1 regulates ZW10 interacting kinetochore protein (ZWINT), thereby promoting tumor progression and spheroid cell formation in CRC [8]. In CRC, individual KIF members exhibit dual functions of fostering and suppressing cancer and only function after forming dimers or binding to cargo [9]. However, the mechanisms of action of these KIF members in the treatment and prognosis of CRC have not been elucidated. For instance, KIF4A and KIF18A may serve as therapeutic targets to improve outcomes in patients with CRC [10,11], but the specific molecular mechanisms by which they enhance patient prognosis remain unclear. Therefore, there is a need for an in-depth study of the KIF members biomarkers associated with the occurrence and development of CRC.

In this study, based on CRC data from The Cancer Genome Atlas (TCGA) and Gene Expression Omnibus (GEO) databases, bioinformatics methods were used to screen biomarkers associated with KIF members, and single-cell sequencing was further utilized to analyze gene expression at the single-cell level to identify subtle differences in cell subpopulations and cell states. This study aimed to identify and analyze CRC biomarkers, which provided a reference for the treatment and prognosis management of CRC. In conclusion, in this study, we hypothesized that the characterization of KIF members could predict survival and immune cell infiltration in patients with CRC and could be expected to be a new target for the future treatment of CRC, providing a new reference for the clinical treatment of patients with CRC.

## 2. Materials and Methods

### 2.1. Data Collection

The transcriptomic data, survival information on patients with CRC, and methylation data from the TCGA-colorectal cancer cohort (COADREAD) (TCGA-CRC), consisting of 616 CRC and 51 control samples, were downloaded from the TCGA database (https://www.cancer.gov/ccg/research/genome-sequencing/tcga, accessed on 22 November 2023) as a test set. Additionally, transcriptomic data (GSE110224 and GSE39582) were gathered from the GEO database (http://www.ncbi.nlm.nih.gov/geo/, accessed on 22 November 2023) as a validation set. Specifically, GSE110224 included 17 CRC and 17 control tissue samples based on the GPL570 platform, and GSE39582 (GPL570) included 585 CRC samples with survival information. Subsequently, 367 CRC samples (including 275 colon cancer (COAD) samples and 92 rectal cancer (READ) samples) and 51 control (including 41 COAD control samples and 10 READ control samples) samples were obtained from the GEPIA2 database (http://gepia2.cancer-pku.cn/, accessed on 22 November 2023). Furthermore, the single-cell RNA sequencing (scRNA-seq) data GSE166555 from the GEO database included 13 CRC and 12 control tissue samples based on the GPL23177 platform. In addition, 43 KIFRGs were mined from the literature [12].

### 2.2. Differential Expression Analysis of the TCGA-CRC Dataset

Differential expression analysis (CRC vs. control) was conducted on the TCGA-CRC dataset via the “DESeq2” package (v 3.44.3) [13], and the screening criteria was |log_2_FoldChange (FC)| > 1 and adj.*p* < 0.05. Volcano maps and heat maps were applied to display the results via “ggplot2” (v 3.3.2) [14] and “pheatmap” (v 0.7.7) [15], respectively.

Furthermore, the methylation data from the TCGA-CRC was statistically analyzed and processed using “methylation Array Analysis” (v 1.26.0) [16]. To explore the genes exhibiting differential methylation status between the CRC and control samples, we obtained methylation data at the methylation CG site level, depicted by beta values, from the TCGA database. Then, the differentially expressed methylation genes (DEMGs) and differential methylation CG sites were separately identified, and the screening conditions were a false discovery rate (FDR) < 0.05 and |Δβ| ≥ 0.1.

### 2.3. Functional Analysis of the Intersecting Genes in the TCGA-CRC Dataset

Intersecting genes were identified by overlapping DEGs and DEMGs. Then, to explore the function of intersecting genes, Gene Ontology (GO) and Kyoto Encyclopedia of Genes and Genomes (KEGG) analyses were employed to reveal the biological processes and signaling pathways of intersecting genes via “clusterProfiler” (v 3.16.0) [17], with a significance threshold of *p* < 0.05.

### 2.4. Survival Analysis of Intersecting Genes in the TCGA-CRC Dataset

To evaluate the relevance of intersecting genes in the survival of patients with CRC from the TCGA-CRC datasets, the patients with CRC were divided into high- and low-expression groups based on the median expression level of the intersecting, and Kaplan–Meier (KM) analysis was performed using the log-rank test (*p* < 0.05). Importantly, genes with significant survival differences were deemed as candidate genes, and the KM curves of candidate genes were plotted.

### 2.5. Identification of Biomarkers in Multiple Datasets

To further evaluate and validate the candidate genes, a rank test was performed to analyze their expression of candidate genes in the TCGA-CRC dataset, GSE110224 dataset, and GEPIA2 database (http://gepia2.cancer-pku.cn/, accessed on 22 November 2023). In addition, we also conducted the survival analysis verification in GSE39582, the patients with CRC from GSE39582 dataset and GEPIA2 database (http://gepia2.cancer-pku.cn/, accessed on 22 November 2023) were divided into high- and low-expression groups according the expression median value of candidate genes, and the KM curves were plotted. Finally, the biomarkers that passed the validation of gene expression and survival analysis were identified.

### 2.6. Gene Set Enrichment Analysis (GSEA) in the TCGA-CRC Dataset

Functional enrichment analysis was performed to further elucidate these biomarkers. First, the CRC samples from the TCGA-CRC were divided into high- and low-expression groups, and the differentially expressed genes (high expression vs. low expression) were acquired and ranked. Additionally, “c2.cp.kegg_legacy.v2023.2. Hs.entrez” from Molecular Signatures Database (MSigDB, https://www.gsea-msigdb.org/gsea/msigdb/index.jsp, accessed on 22 November 2023) was selected as background gene set. Subsequently, a GSEA enrichment analysis was conducted using “clusterProfiler”, the screening condition was *p* < 0.05.

### 2.7. Immune Infiltration and Immune Checkpoint Analysis in the TCGA-CRC Dataset

The single-sample GSEA algorithm was used to calculate the infiltration score of 28 immune cells in the TCGA-CRC samples. The difference in infiltration scores between the CRC and control samples was analyzed via the Wilcox test (*p* < 0.05). In addition, Spearman analysis was applied to explore the correlation between biomarkers and differential immune cells by “corrplot” (v 0.92) [18], and the threshold was set to |r| > 0.30 and *p* < 0.05.

Furthermore, immune checkpoint inhibitors play a crucial role in immune function and have various clinical implications for immunotherapy. In this study, the following common immune checkpoints were selected: IDO1, PD-L1 (CD274), PD-L2 (PDCD1LG2), TIM-3 (HAVCR2), TIGIT, PD-1 (PDCD1), LAG3, ICOS, and CD27. The expression of common immune checkpoints between the CRC and control samples was analyzed (*p* < 0.05). Subsequently, Spearman analysis was applied to reveal the correlation between biomarkers and common immune checkpoints, and the threshold was set to |r| > 0.30 and *p* < 0.05.

### 2.8. Regulation Network Analysis

To further explore the potential mechanisms of the biomarkers, transcription factor (TF)-targeting biomarkers were predicted using the ChEA3 database (https://amp.pharm.mssm.edu/chea3/, accessed on 22 November 2023). Then, a TFs-mRNA network was constructed and visualized using Cytoscape (v. 3.9.1) [19]. Additionally, the differentially expressed miRNAs (DEmi) (CRC vs. control) were identified by “DESeq2” in the TCGA-CRC, with a threshold of adj. *p* < 0.05, and |log_2_FC| > 1. Later, miRNAs targeting biomarkers were predicted using the miRWalk Website (http://mirwalk.umm.uni-heidelberg.de/, accessed on 22 November 2023). The key miRNAs were immediately acquired by overlapping the downregulated DEmi and predicted miRNAs. Next, an miRNA-mRNA network was constructed. Subsequently, by organizing the obtained TF mRNA and miRNA mRNA, a TF-mRNA-miRNA regulation network was constructed.

### 2.9. Drug Sensitivity Analysis in the TCGA-CRC Dataset

To assess the sensitivity of patients with CRC to conventional drugs, the patients with CRC in the TCGA-CRC were first divided into high- and low-expression groups based on the median expression value of the biomarker, and 138 drugs were acquired from the Cancer Genome Project (CGP, https://cancer.sanger.ac.uk/, accessed on 22 November 2023) database. Next, the inhibitory concentration 50 (IC_50_) value of each drug for the high- and low-expression groups was calculated using “pRRopheticPredict” (version 0.5). In addition, the Wilcoxon test was conducted to compare the differential expression of the IC_50_ for each drug between the high- and low-expression groups (*p* < 0.05), and the top four drugs with significantly differential expression are presented.

### 2.10. Single-Cell RNA Sequencing (scRNA-Seq) Analysis

The samples from the GSE166555 dataset were integrated for the scRNA-seq analysis by “Seurat” (v 3.1.5) [20], and the parameter was set to min.cells = 3 and min.features = 200. First, the low-quality cells were removed according to the quality control (QC) criteria, which included a library size greater than 500 and less than the 95th percentile (i.e., fewer than 10,000 cells); gene count less than the 95th percentile; mitochondrial content less than 10%; and scDblFinder determined as non-doublet cells by the detected from “scDblFinder” (v 1.12.0) [21]. After the data were unified, the top 2000 highly variable genes (HVGs) were selected by “FindVariableFeatures”. Next, based on these 2000 HVGs, principal component analysis (PCA) was performed, and the principal components (PCs) were determined. Subsequently, the cells were clustered using t-distributed stochastic neighbor embedding (t-SNE). Subsequently, the cells were annotated based on the marker genes from the CellMarker database (http://biocc.hrbmu.edu.cn/CellMarker/, accessed on 22 November 2023).

### 2.11. Differential Expression Analysis, GSEA, and Biomarkers Expression Analysis in Annotated Cells

Furthermore, the DEGs between the CRC and control samples in annotated cells were identified, and the screening standard was an |average log_2_FC| > 0.25, pct > 0.1, and adj.p < 0.05. To further explore the function of DEGs in annotated cells, the log_2_FC of each gene was ranked from highest to lowest. Subsequently, the GSEA of DEGs in the annotated cells was performed via “clusterProfiler”, and the threshold was |NES| > 1 and NOM *p* < 0.05. In addition, the expression of biomarkers in the annotated cells was analyzed and visualized.

### 2.12. Pseudo-Temporal Analysis and Single-Cell Regulatory Network Inference and Clustering (SCENIC)

To explore cell differentiation states and directions, pseudo-temporal analysis was performed using “Monocle” (v 2.26.0) [22]. Later, to explore the upstream regulators in annotated cells, the RcisTarget and GRNBoost databases were employed to perform SCENIC analysis. Additionally, the “RcisTarget” (v 1.19.2) [23] was applied to identify TFs with overexpression binding motifs in a gene list. Then, “AUCell” was employed to calculate the Regulatory Specificity Score (RSS) for each cell. The TFs were displayed using a bubble map and heat map.

### 2.13. Experimental Validation Analysis of Target Genes

The CRC and normal tissue samples were collected from 25 patients with CRC who underwent surgery at the Guangdong Second Provincial General Hospital. This study was approved by the Ethics Committee of the Institution (approval no: 2024-KY-KZ-012-02). A written informed consent form was obtained from each patient. Samples of both tumor and normal tissues were harvested from surgically removed tumors and the distal normal intestinal mucosal epithelium, respectively. Following the HE staining, DNA was extracted from the tumor regions and distal normal intestinal mucosal areas. DNA extraction was performed according to the protocol provided in the DNA extraction kit (TianGen Biochemistry, Beijing, China).

Real-time quantitative PCR (qPCR) was performed using the TaqMan probe method in 96-well plates. The extracted DNA was combined with probes, primers, and reaction mix. Each well contained target genes labeled with FAM probes and reference genes labeled with CY5 probes. The reaction mixture in each well consisted of 2 µL of tissue DNA at a concentration of 5 ng/µL, 5 µL of the mix, and 0.64 µL of primers and probes. The PCR conditions were set to one cycle of 5 min at 95 °C, followed by 20 cycles of 15 s at 95 °C, and 30 s at 64 °C, with fluorescence detection at 60 °C for 10 s over 40 cycles. The relative expression of the target gene was calculated using the 2^−△△Ct^ method.

### 2.14. Statistical Analysis

All statistical analyses were performed using R (version 4.2.3). A *p*-value < 0.05 was used, unless otherwise stated.

## 3. Results

### 3.1. The 12 Intersecting Genes Were Mainly Enriched in Microtubule Related Function

Differential expression analysis identified 12,479 DEGs (7736 upregulated and 4743 downregulated) between the CRC and control samples in the TCGA-CRC (Figure 1A,B). In addition, no outliers were detected in the TCGA-CRC (Figure 1C). Furthermore, Figure 1D shows that all samples exhibited a good fit, indicating minimal or negligible batch effects between the samples. These findings confirmed that the data quality was sufficient to support subsequent analyses. Subsequently, 11,319 DEMGs (CRC vs. control) and 84,662 differentially methylated CG sites (37,936 hypermethylated sites and 46,726 hypomethylated sites) were identified separately (Figure 1E).

Subsequently, 12 intersecting genes were identified by overlapping 12,479 DEGs, 11,319 DEMGs, and 43 KIFRGs (Figure 2A). In addition, enrichment analysis of 12 intersecting genes showed that they were enriched in 214 GO entries and 11 KEGG pathways. In detail, “axonal transport”, “microtubule-associated complex”, and “microtubule motor activity”, etc. GO terms were also identified (Figure 2B). Additionally, for the KEGG analysis, 12 intersecting genes were enriched in “dopamine nerve synapses”, “endocytosis”, “non-small cell lung cancer”, etc. (Figure 2C).

### 3.2. KIFC3 Were Identified as Biomarkers

Based on the 12 intersecting genes, we found that *KIFC2* and *KIFC3* had significant survival differences (*p* < 0.05) (Table 1). Meanwhile, the KM curves of *KIFC2* (*p* = 0.0436) and *KIFC3* (*p* = 0.0199) showed that low-expression groups in the TCGA-CRC had a significantly higher survival probability (Figure 3A,B).

Furthermore, the expression analyses of *KIFC2* and *KIFC3* revealed that *KIFC3* exhibited significantly higher expression levels in the CRC samples in both the TCGA-CRC and GSE110224 datasets. In contrast, *KIFC2*’s expression was significantly elevated only in the CRC samples from the TCGA-CRC dataset (Figure 3C,D). In addition, *KIFC3* had a significant survival difference between the high- and low-expression groups in the GSE39582 dataset, with the *KIFC3* low-expression group having a significantly higher survival rate than the *KIFC3* high-expression group; however, there was no significant survival difference between the *KIFC2* high- and low-expression groups in the GSE39582 dataset (Figure 3E,F). Subsequently, in the GEPIA2 database, the expression level of *KIFC2* in rectal cancer (READ) was significantly lower than that of the control group, while the expression level of *KIFC3* in colon cancer (COAD) was significantly higher than that of the control group (*p* < 0.05) (Figure 3G,H). The survival rate of the *KIFC3* low-expression group was significantly higher than that of the *KIFC3* high-expression group (*p* = 0.0091) (Figure 3I), whereas there was no significant difference in survival rate between the *KIFC2* high- and low-expression groups (Figure 3J). In summary, *KIFC3* was associated with higher survival rates, *KIFC3* was identified as a potential biomarker.

Additionally, based on the biomarker (*KIFC3*), the GSEA showed 88 KEGG pathways were detected, such as “Ecm receptor intersecting”, “chemokine signaling pathway”, and “pathways in cancer” (Figure 3K).

### 3.3. KIFC3 Had a Positive Correlation with Differential Immune Cells and Common Checkpoints

Twenty-four differential immune cells were detected in 28 immune cells, included Natural killer (NK) cells, macrophages, and eosinophils (*p* < 0.05) (Figure 4A). In addition, we found that *KIFC3* was significantly positively correlated with NK cells (r = 0.455, *p* < 0.05), NK T cells (r = 0.411, *p* < 0.05), and central memory CD8 T cells (r = 0.409, *p* < 0.05) (Figure 4B).

Moreover, among the nine common immune checkpoints, excluding IDO1, the remaining eight immune checkpoints exhibited a significant difference between the CRC and control samples (*p* < 0.05) (Figure 4C). In addition, there was a significant correlation between *KIFC3* and nine immune checkpoints (|r| > 0.3, *p* < 0.05) (Figure 4D).

### 3.4. Regulatory Network of KIFC3 in CRC

By predicting, 79 TFs targeting *KIFC3* were acquired, and a TF-mRNA network (80 nodes and 79 edges) was constructed, such as “TCF3-*KIFC3*” and “ELK1-*KIFC3*” (Figure 5A).

In addition, 506 DEmis (CRC vs. control) were detected, of which 167 were upregulated and 339 were downregulated (Figure 5B,C). Using miRWalk, 323 miRNAs were identified. Following this, 12 key miRNAs were selected by overlapping 339 downregulated miRNAs and 323 predicted miRNAs (Figure 5D). An miRNA-mRNA network comprising 13 nodes and 12 edges was constructed (Figure 5E), such as “hsa-miR-4741”-*KIFC3*, “hsa-miR-151b”-*KIFC3*, etc.

After integrating the above interaction relationships, a TF-mRNA-miRNA (1 mRNA, 12 miRNAs, 79 TFs, and 91 interaction relationships) was built, such as TET1-*KIFC3*-“hsa-miR-572” and ELF1-*KIFC3*-“hsa-miR-5193” (Figure 5F).

### 3.5. AZD.2281, Nilotinib, PD.173074, and Shikonin Were Helpful for Treating CRC

According to the drug sensitivity analysis, a total of 120 drugs exhibited significant differences between the high- and low-expression groups, with the IC_50_ values of AZD.2281, nilotinib, PD.173074, and shikonin being significantly lower in the high-expression group for *KIFC3*, indicating that *KIFC3* was more sensitive to these drugs in the high-expression group (Figure 6).

### 3.6. A Total of 16 Cell Clusters Were Annotated

Initially, ineligible cells were filtered out, and the remaining cells and genes were used for further analysis. Subsequently, 2000 HVGs were detected and the top 10 genes were annotated (Figure 7A). PCA was performed, and the top 20 PCs were selected for subsequent analyses (Figure 7B). Later, the cells were classified into 35 distinct cell clusters using t-SNE dimensionality reduction (Figure 7C), and the distribution of cell clusters in the CRC and control samples is presented in Figure 7D. After annotation, 16 cell types were annotated, including goblet cells, naive CD4^+^ T cells, plasma B cells, epithelial cells, follicular B cells, enterocytes, CD8^+^ Tem, CD4 Treg, M1-like macrophages, fibroblast cells, NKT, M2-like macrophages, endothelial cells, mast cells, dendritic cells, and glial cells (Figure 7E,F).

### 3.7. GSEA Revealed 16 Cell Clusters Involved in Pathways, Including Antigen Processing and Presentation, Among Others

DEGs were identified in 16 cell clusters. A GSEA was then performed. We found CD4 Treg, CD8^+^ Tem, NKT, naive CD4^+^ T cell, mast cell, and dendritic cells were mainly enriched in “antigen processing and presentation”, “toxoplasmosis”, etc. Endothelial cells and enterocytes were mainly enriched in “PPAR signaling pathway”, epithelial cells and M2-like macrophages were enriched in “rheumatoid arthritis”, goblet cells, follicular B cells, and M1-like macrophages were enriched in “IL−17 signaling pathway” and “lipid and atherosclerosis”, fibroblast cells were enriched in “TNF signaling pathway”, glial cells were involved in “*Staphylococcus aureus* infection”, and plasma B cells were enriched in “human papillomavirus infection”, “fluid shear stress and atherosclerosis”, etc. (Appendix A).

### 3.8. Macrophages Play a Critical Role in the Development of CRC

Expression analysis of *KIFC3* revealed that *KIFC3* was expressed in enterocyte cell, M1-like macrophages, M2-like macrophages, and glial cells. Meanwhile, in these cells, the expression almost exclusively occurred in tumor samples compared to control samples, indicating that *KIFC3* plays an important role in the treatment of CRC (Figure 8A).

In addition, owing to the critical roles of macrophages, in this study, M1- and M2-like macrophages were selected to perform a pseudo-temporal analysis. The results show that the differentiation direction was from left to right (from dark blue to light blue), and the cells were divided into five states (stage 1–5) and three branches (Figure 8B–D). Additionally, we observed that most M1-like macrophages were present in the early stages of differentiation, whereas M2-like macrophages were predominant in the later stages of differentiation (Figure 8E). Moreover, *KIFC3* was highly expressed in the CRC tumor samples (Figure 8F). Similarly, *KIFC3* was highly expression in M1-like macrophages (Figure 8G).

### 3.9. MAFB Was Key TF for M2-like Macrophages

By searching for TFs, we found that MAFB had a stronger interaction with M2-like macrophages, NFE2 had a strong intersection with M1-like macrophages, and EOMES and CD8+ Tem had strong intersections (Figure 9A). The heat map also showed the same results (Figure 9B).

### 3.10. Expression of KIFC3in CRC Tissues and Normal Tissues in qPCR

In 25 CRC tissue samples, the expression level of *KIFC3* was significantly lower than that of the corresponding normal tissues (*p* < 0.05) (Figure 10).

## 4. Discussion

CRC is a common malignant tumor of the digestive system, and its incidence has increased significantly in recent years, becoming a global health concern [1]. KIF members, an intracellular transport molecule, plays an important role in a variety of malignant tumors, including CRC [9]. This study used bioinformatics methods to screen out the CRC-related biomarker *KIFC3* from the TCGA and GEO databases and further explored its biological effects and potential mechanisms.

*KIFC3* encodes a microtubule-associated protein responsible for regulating microtubule-dependent transport and distribution within cells and plays a crucial role in maintaining cellular structure and function [24]. Previous studies have shown that *KIFC3* overexpression is associated with various tumors. In HCC, high expression of *KIFC3* is associated with poor overall survival and promotes HCC progression through PI3K/AKT/mTOR signaling [25], whereas upregulated *KIFC3* expression in GC correlates with advanced T stage, poor prognosis in patients, and promotes cancer progression and metastasis by activating the Notch1 pathway [26]. *KIFC3* expression levels were positively correlated with the proliferation, migration, and invasion abilities of CRC cells and enhanced the EMT process through the PI3K/AKT/mTOR pathway, suggesting that *KIFC3* may be involved in the regulation of CRC progression [27]. Furthermore, the high expression of *KIFC3* in the low-risk group and its association with high survival in patients with CRC suggested that it might be a potential therapeutic target for CRC treatment. The correlation between differential immune cells and checkpoint inhibitors also indicated that targeting *KIFC3* could potentially enhance the immune response against CRC. Overall, these findings suggest that *KIFC3* plays a crucial role in CRC and may serve as a promising biomarker and therapeutic target for the disease. Further research is needed to fully understand the mechanisms underlying the role of *KIFC3* in CRC and develop targeted therapies that can effectively inhibit its function.

GSEA enrichment analysis was used to find that the main pathways in which *KIFC3* is enriched include “ECM receptor intersection”, “chemokine signaling pathways”, and “pathways in cancer”. These insights suggest that *KIFC3* plays a crucial role in these biological processes and could be pivotal for understanding its function and potential as a therapeutic target in CRC. Studies have reported that MiR-215-5p can reduce liver metastasis of CRC by affecting the ECM receptor intersection pathway [28], suggesting that this miRNA may be a promising target for future treatment strategies for patients with CRC and that the Chemokine signaling pathway regulates cell migration and positioning by transmitting signals between cells through receptors [29]. In osteosarcoma metastasis, genes associated with chemokine activity, chemokine receptor binding, and the TNF signaling pathway are enriched, and *KIFC3* has been identified as a significant prognostic biomarker [30]. In the immune cell analysis, we identified disparities in 24 types of immune cells, including NK cells, macrophages, and eosinophils. Over the years, studies have focused on the impact of NK cell function on tumors. In CRC, the expression of NKp44 and NKp46 is reduced in CD56dim NK cells and NKT-like cells [31]. Cytotoxic lymphocytes, including CD8-T cells and NK cells, are associated with improved overall survival in CRC and are relevant prognostic factors [32]. Enhancing our understanding of the interplay between NK cells and CRC could expand therapeutic and prognostic strategies for addressing this disease. In addition, blood eosinophil count is inversely related to CRC risk [33]. Some studies have also found that the expression of *KIFC3* during osteosarcoma metastasis is positively correlated with follicular helper T cells, suggesting that *KIFC3* may be involved in immune responses during osteosarcoma metastasis [30]. Our study revealed that the expression of *KIFC3* was significantly positively correlated with NK cells and eosinophils, indicating that *KIFC3* may play a role in immune activities during the development of CRC.

*KIFC3*-based research on various immune cells can help in developing more effective immunotherapy strategies. We identified 120 sensitive drugs, including AZD.2281, nilotinib, PD.173074, and shikonin. Their low IC50 values in the *KIFC3* high-expression group suggests heightened sensitivity, which is indicative of altered gene expression patterns. This analysis implies that since *KIFC3* is highly expressed in disease samples, these drugs could potentially be effective in treating CRC. AZD.2281 is a PARP inhibitor that interferes with the DNA repair process by inhibiting PARP [34]. It is currently mainly used to treat breast [35] and ovarian cancers [36] related to BRCA1/2 mutations. Clinical trials to further validate the efficacy and safety of AZD.2281 in the treatment of CRC are imperative, as well as to ascertain the specific patient cohorts that may derive optimal benefit from this therapeutic approach. Nilotinib is a small-molecule targeted drug that is a tyrosine kinase inhibitor [37]. In vitro experiments have shown that nilotinib targeting DDR1 kinase activity may be beneficial in patients with CRC [38]. While nilotinib does not affect NK cell cytotoxicity, it impairs NK cell cytokine production at high concentrations [39]. Additionally, studies have observed that nilotinib strongly inhibits the function of CD8+ T lymphocytes [40]. Therefore, we speculate that nilotinib may exert its therapeutic effect on patients with CRC by modulating the functions of NK cells or T cells. We report for the first time that PD.173074 may have the potential to treat CRC. Studies have shown that patients with CRC with high inflammation are more sensitive to 5-Fluorouracil (5-FU), camptothecin, irinotecan, and docetaxel, while those with low inflammation respond better to non-traditional drugs, such as lapatinib, selumetinib, vorinostat, and gefitinib [41]. These results further confirm the heterogeneity among different cohorts, indicating that disease risk may lead to varied responses to drug treatment.

The emergence of single-cell analysis technology provides an important means for in-depth understanding of the tumor microenvironment. In tumor research, single-cell analysis can more comprehensively analyze the distribution and function of various cell subtypes within tumors, thus providing a more accurate basis for precision medicine [42]. A quasi-chronological analysis revealed a notable increase in both M1 and M2 macrophages in samples from patients with CRC compared with normal samples, with a clear trend of M1 macrophages transitioning into M2 macrophages over time. In early inflammation, macrophages are activated by various stimuli, primarily polarizing toward the M1 type, to combat pathogenic bacteria. During inflammation resolution, macrophages exhibit M2 characteristics by clearing apoptotic neutrophils via efferocytosis [43]. In addition, as tumors progress, M1 gradually transforms into the M2 phenotype, promoting tumor growth and metastasis [44]. We speculate that *KIFC3* is specifically overexpressed in M1/M2 macrophages, and macrophage polarization is closely associated with the formation of an immunosuppressive tumor microenvironment. Low expression of *KIFC3* may inhibit the transition of macrophages to a pro-cancer phenotype, thereby improving CRC prognosis. Based on the above analysis, we speculate that the differentiation and transformation of M1 and M2 macrophages in CRC may be important mechanisms in the process of tumor progression. This polarized transformation of macrophages may induce immunosuppression and tumor escape from the tumor microenvironment. This discovery provides new ideas for the development of immunotherapy strategies targeting macrophages and is expected to bring new breakthroughs in the treatment of patients with CRC.

In this study, *KIFC3* was identified as a biomarker associated with CRC through comprehensive screening, differential analysis, survival analysis, and gene expression verification. The GSEA enrichment analysis, immune-related analysis, and drug sensitivity analysis were conducted to investigate biological connections. In addition, single-cell analysis to reveal CRC’s mechanisms at the cellular level provides new insights into disease pathogenesis and personalized treatment strategies. However, this study relied on public sources of data for the initial exploration and may suffer from an inadequate sample size and lack of experimental validation. Therefore, we will knock down *KIFC3* by CRISPR/Cas9 or siRNA, or overexpress *KIFC3* by plasmid, and observe its effects on CRC cell proliferation, migration, invasion, immune cell function or related pathway interactions. The effects of *KIFC3* on the CRC cell phenotype will be evaluated using CCK-8, EdU staining, and Transwell assay. Immunoprecipitation (Co-IP) or protein interaction assays will be used to verify whether *KIFC3* directly interacts with immune-cell-associated proteins.

## 5. Conclusions

In this study, we first identified *KIFC3* as a novel CRC biomarker. *KIFC3* was found to be associated with ECM receptor interaction, chemokine signaling, and cancer pathways. Molecularly, it binds to TCF3, ELK1, hsa-miR-4741, and hsa-miR-151b, offering new insights into CRC pathogenesis. Notably, our work on *KIFC3*–immune-cell interactions revealed its role in the immune microenvironment, indicating that *KIFC3* is a potential immunotherapy target. This discovery paves the way for precision treatment of CRC and new clinical strategies.

## Figures and Tables

**Figure 1 biomedicines-13-00859-f001:**
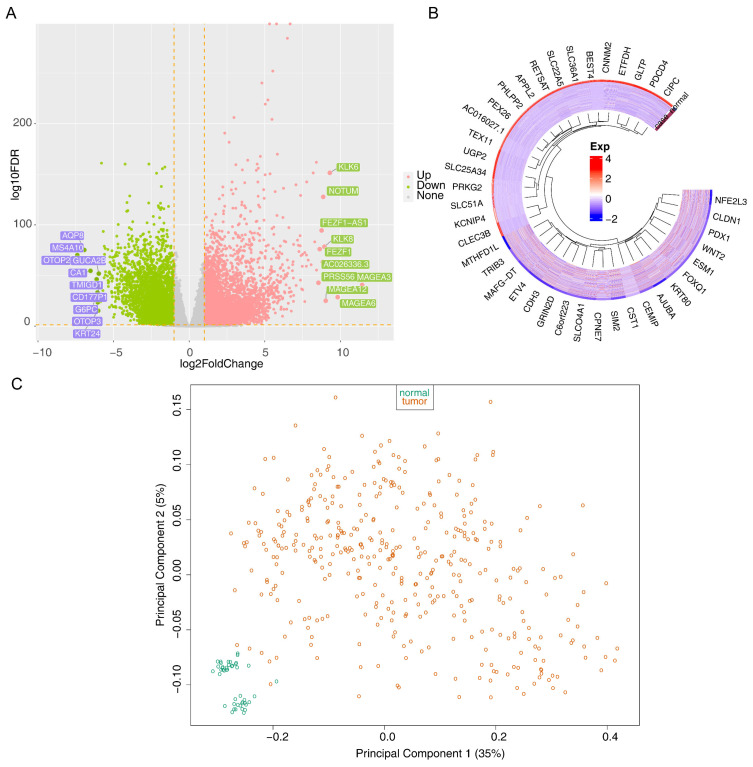
The identified intersecting genes: (**A**,**B**) plots of up- and downregulated differentially expressed genes between the colorectal cancer (CRC) and control samples. In the (**A**) plot, pink dots represented upregulated genes, and green dots represented downregulated genes; (**C**) Outlier sample detection (Orange dots represented tumor group samples, and green dots represented control samples); (**D**) Sample fit (Orange lines represented the tumor group, and green lines represented the normal group); (**E**) volcano plot of differentially expressed methylated genes and methylation sites between the CRC and control samples (Red dots represented upregulated genes, and blue dots represented downregulated genes).

**Figure 2 biomedicines-13-00859-f002:**
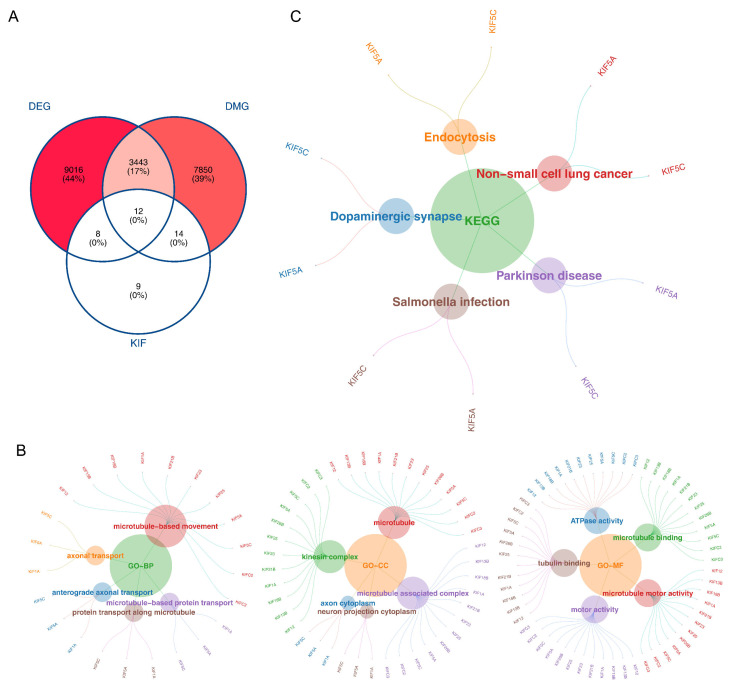
The 12 intersecting genes identified**:** (**A**) Venn diagram of the intersecting genes; (**B**) Gene Ontology (GO) analysis; (**C**) Kyoto Encyclopedia of Genes and Genomes (KEGG) analysis.

**Figure 3 biomedicines-13-00859-f003:**
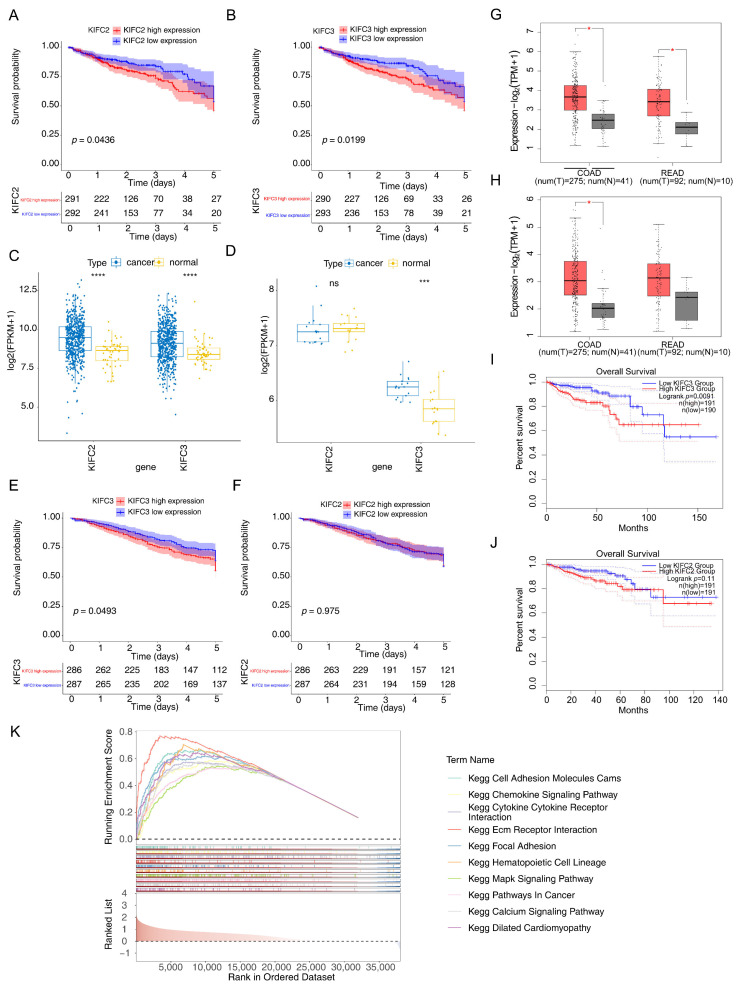
The identified biomarkers: (**A**,**B**) KM curves for the *KIFC2* and *KIFC3* survival analysis; (**C**,**D**) expression analysis of *KIFC2* and *KIFC3* in the CRC datasets; ns represented no significance, *** represented *p* < 0.001 and **** represented *p* < 0.0001. (**E**,**F**) KM curves of the *KIFC2* and *KIFC3* high- and low-expression groups in the GSE39582 dataset; (**G**,**H**) expression analysis of *KIFC2* and *KIFC3* in the GEPIA2 database; * represented *p* < 0.05. (**I**,**J**) KM curves of the *KIFC3* and *KIFC2* high- and low-expression groups in the GEPIA2 database; (**K**) GSEA result based on the *KIFC3* biomarker.

**Figure 4 biomedicines-13-00859-f004:**
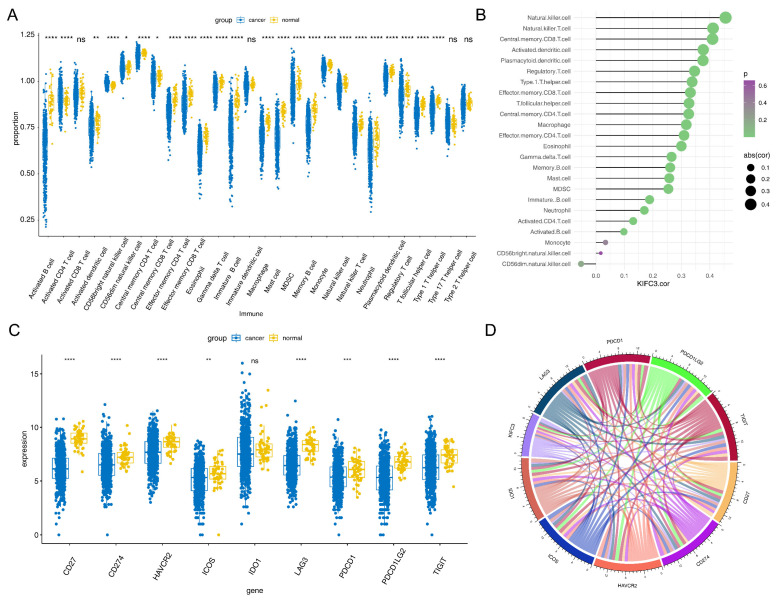
The biomarkers immune cells analysis: (**A**) differential immune cells analysis (ns represented no significance, * represented *p* < 0.05, ** represented *p* < 0.01 and **** represented *p* < 0.0001); (**B**) correlation analysis between Kinesin family member C3 (*KIFC3*) and immune cells; (**C**) differential expression of common immune checkpoints (ns represented no significance, ** represented *p* < 0.01, *** represented *p* < 0.001 and **** represented *p* < 0.0001); (**D**) correlation analysis between *KIFC3* and immune checkpoints.

**Figure 5 biomedicines-13-00859-f005:**
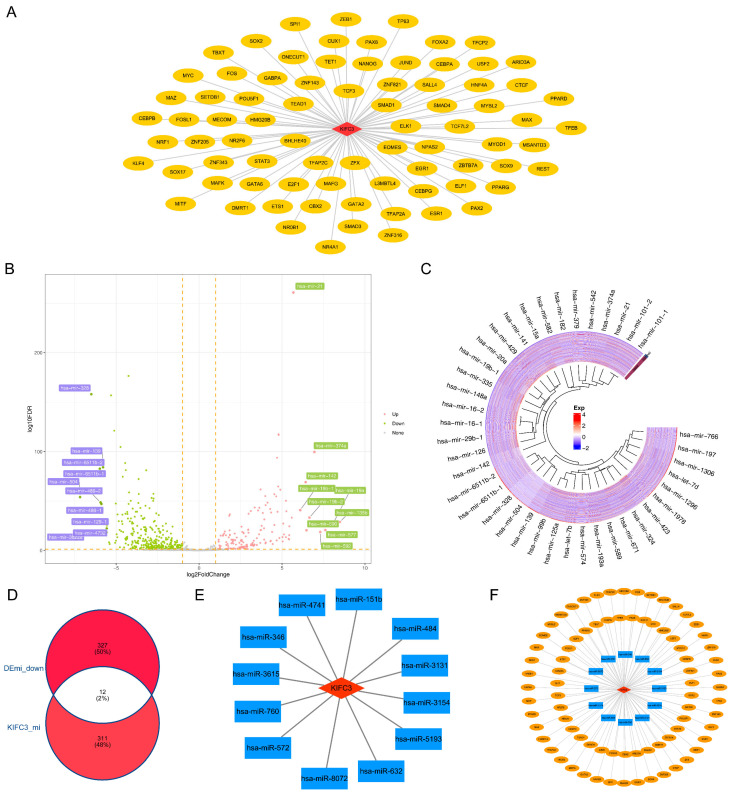
The constructed regulatory networks: (**A**) TF-mRNA network targeting *KIFC3* (The orange ellipse represented transcription factors); (**B**,**C**) analysis of DEmis up- and downregulation between the CRC and control samples (Pink dots represented upward adjustments, while green dots represented downward adjustments); (**D**) selection of key miRNAs; (**E**) miRNA-mRNA network (The blue square represents miRNA); (**F**) integrated TF-mRNA-miRNA network.

**Figure 6 biomedicines-13-00859-f006:**
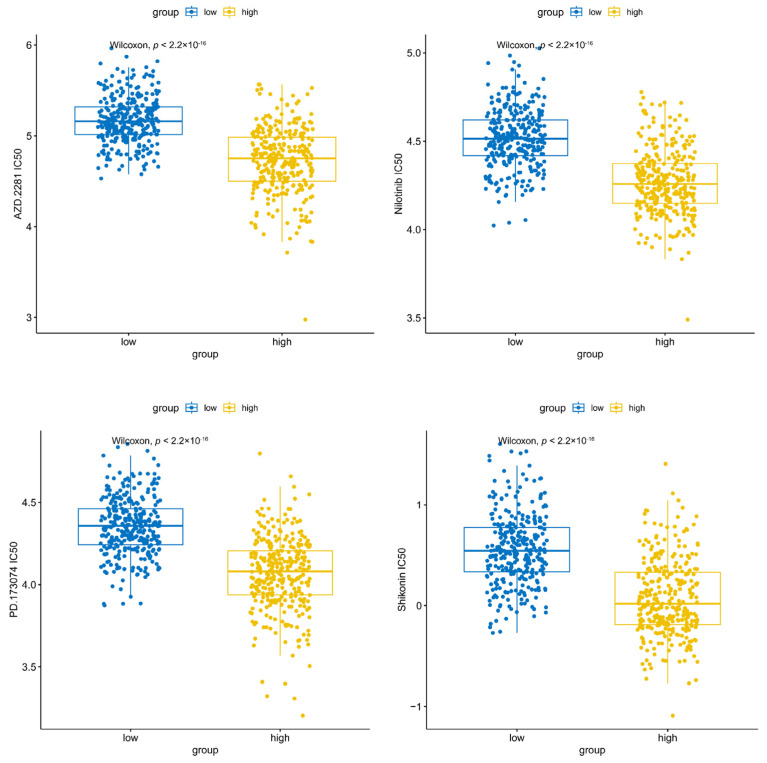
Drug sensitivity analysis between the *KIFC3* high-expression group and low-expression group.

**Figure 7 biomedicines-13-00859-f007:**
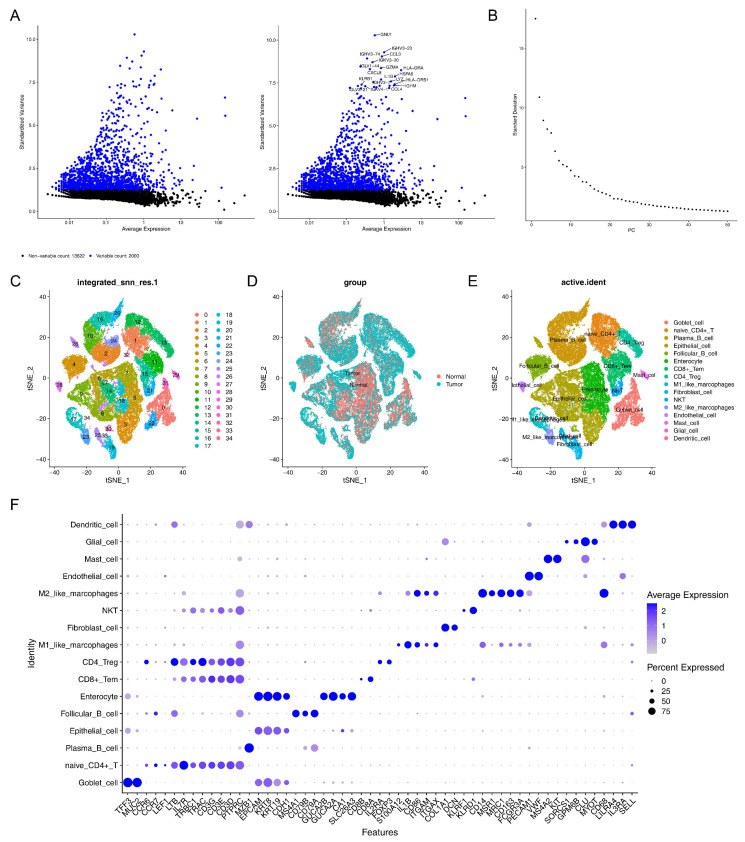
The cell clusters annotated: (**A**) annotation of the top 10 highly variable genes; (**B**) principal component analysis (PCA); (**C**) T-distributed stochastic neighbor embedding (t-SNE) dimensionality reduction for cell clustering; (**D**) distribution of cell clusters in the CRC and control samples; (**E**,**F**) annotation of 16 cell clusters.

**Figure 8 biomedicines-13-00859-f008:**
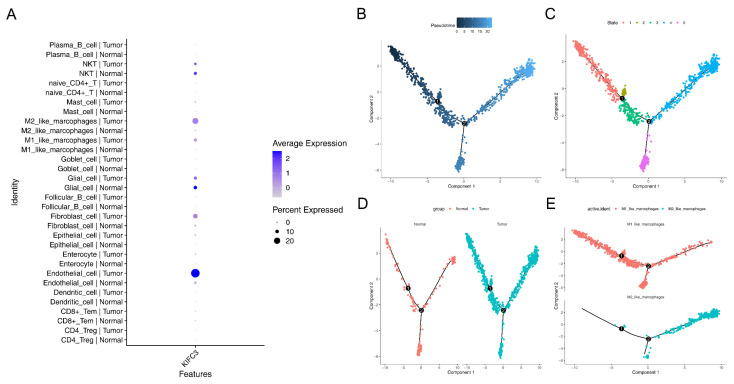
The expression analysis of *KIFC3*: (**A**) expression analysis of *KIFC3* in different cell types; (**B**–**D**) pseudo-temporal analysis of the M1- and M2-like macrophages; (**E**) distribution of the M1- and M2-like macrophages in differentiation stages; (**F**) high expression of *KIFC3* in CRC tumor samples; (**G**) high expression of *KIFC3* in M1-like macrophages.

**Figure 9 biomedicines-13-00859-f009:**
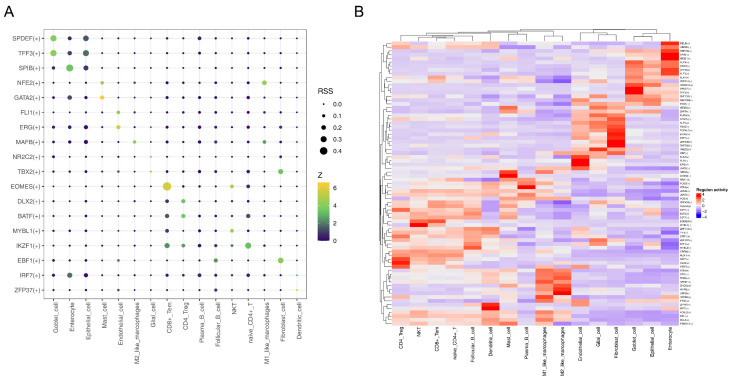
The TFs’ analysis: (**A**) interaction of TFs with macrophage subtypes; (**B**) heat map visualization of the TF interactions.

**Figure 10 biomedicines-13-00859-f010:**
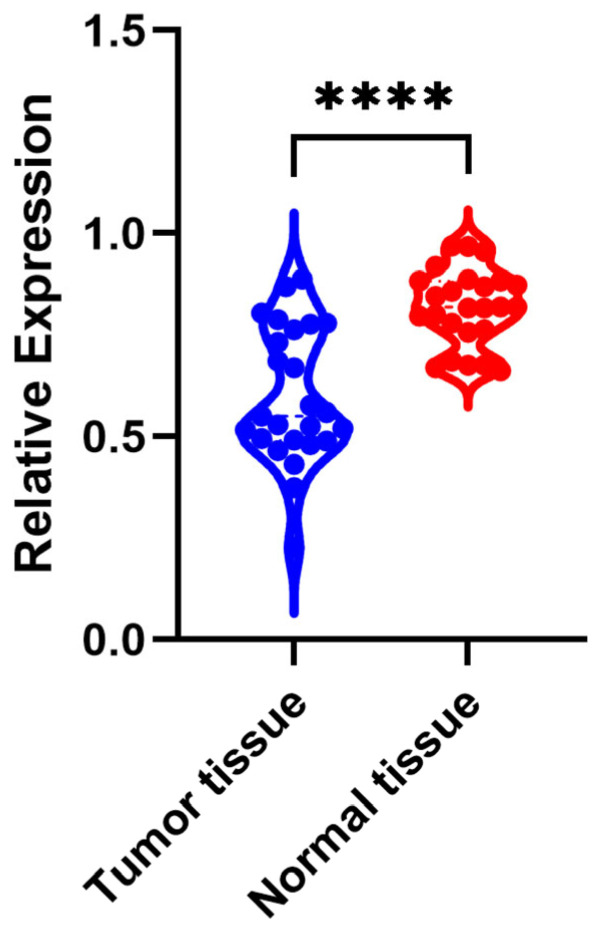
QPCR analysis of *KIFC3*’s expression in CRC and normal tissue. Blue dots represented KIFC3 expression in tumor tissue, and red dots represented KIFC3 expression in normal tissue. **** represented *p* < 0.0001.

**Table 1 biomedicines-13-00859-t001:** Differential survival *p*-values of the KIF member-related methylation genes.

Gene	*p*-Value
*KIF12*	0.796061026
*KIF13B*	0.277954542
*KIF18B*	0.505774554
*KIF1A*	0.842258936
*KIF21B*	0.778932753
*KIF23*	0.85609048
*KIF25*	0.805087841
*KIF26B*	0.112269083
*KIF5A*	0.377814398
*KIF5C*	0.340239373
*KIFC2*	0.043597233
*KIFC3*	0.019886808

## Data Availability

The datasets used and/or analyzed during the current study are available from the corresponding author upon reasonable request.

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
