# Peer review of "Identification of KIFC3 as a Colorectal Cancer Biomarker and Its Regulatory Mechanism in the Immune Microenvironment Based on Integrated Analysis of Multi-Omics Databases"

_biomedicines, 2025, doi:10.3390/biomedicines13040859_

Round 1

Reviewer 1 Report

Comments and Suggestions for Authors

The article covered the identification of KIFC3 as a CRC biomarker and showed the correlation of immune cells in this aspect. The study shows that inhibition of KIFC3 could improve immunotherapy efficacy and provide candidate drugs (e.g., AZD2281, Nilotinib, PD.173074) with therapeutic promise against CRC. I have a few concerns here about the manuscript.

  • First, the paper appears to be bioinformatics only, purely relying on multi-omics data analysis. There is no in vitro or in vivo experimental validation. How are you going to justify this?
  • While KIFC3 is implicated in CRC formation and interactions with immune cells, causality has not been established. It should be utilized through functional assays to determine whether KIFC3 directly impacts such pathways. 
  • Another example, the study shows a role for KIFC3 in M1-to-M2 macrophage conversion but possesses no mechanistic information to state this. further experiments such as co-culture experiments using immune cells to determine whether KIFC3 has an effect on macrophage polarization or NK cell function.
  • The study employs TCGA and GEO data, but that is not representative of heterogeneous cohorts of patientsValidation in separate cohorts should be done, pls check on that
  • Some of the sentences are lengthy and can be condensed for greater clarity in the introduction plus discussion. also i see overgeneralization of findings such as  "providing strong theoretical foundations" and "presenting new methods to clinical CRC treatmentoverstate the immediate clinical applicability without experimental confirmation. may be more strong references. 
  • i don't see future direction mentioned in the paper
  • the introduction does not clarify how this research novelty differs from previous work on KIFs in cancer.

Author Response

The article covered the identification of KIFC3 as a CRC biomarker and showed the correlation of immune cells in this aspect. The study shows that inhibition of KIFC3 could improve immunotherapy efficacy and provide candidate drugs (e.g., AZD2281, Nilotinib, PD.173074) with therapeutic promise against CRC. I have a few concerns here about the manuscript.

First, the paper appears to be bioinformatics only, purely relying on multi-omics data analysis. There is no in vitro or in vivo experimental validation. How are you going to justify this?

Re: Thank the reviewer for the comments. Indeed, we have verified the results of this study through PCR experiments, but we acknowledge that further experimental validation is still needed. Therefore, we have added the final section of the discussion (line 498-505) to emphasize the preliminary nature of the mechanisms explored in this study and the need for subsequent verification. Thank you again for your insightful comments, which are crucial to the advancement of our research.

While KIFC3 is implicated in CRC formation and interactions with immune cells, causality has not been established. It should be utilized through functional assays to determine whether KIFC3 directly impacts such pathways.

Re: Thank the reviewer for the comments. We acknowledge the limitations due to the lack of experimental validation (lines 498-500). Based on your suggestions, we have further discussed the role of KIFC3 in CRC in (line 439-444) of the discussion. Thank you again for your thoughtful review.

Another example, the study shows a role for KIFC3 in M1-to-M2 macrophage conversion but possesses no mechanistic information to state this. further experiments such as co-culture experiments using immune cells to determine whether KIFC3 has an effect on macrophage polarization or NK cell function.

The study employs TCGA and GEO data, but that is not representative of heterogeneous cohorts of patients. Validation in separate cohorts should be done, pls check on that

Re: We feel great thanks for your professional review work on our article. Based on your suggestions, we have added the expression levels and survival differences of KIFC2 and KIFC3 in the GEPIA2 database to the Materials and Methods (section 2.5, lines 132-134) and Results (section 3.2, lines 282-289). Additionally, we have discussed the heterogeneity between different cohorts in the Discussion section (line 465-467), noting that this heterogeneity may be influenced by disease risk. Thank you again for your thorough review of this study, and we look forward to receiving your further feedback.

Some of the sentences are lengthy and can be condensed for greater clarity in the introduction plus discussion. also i see overgeneralization of findings such as "providing strong theoretical foundations" and "presenting new methods to clinical CRC treatment" overstate the immediate clinical applicability without experimental confirmation. may be more strong references.

Re: Thank you very much for reviewing our study and providing valuable feedback. Based on your suggestions, we have simplified the concluding summary in the introduction (lines 80-89) and removed the corresponding results. In the conclusion (lines 507-513), we now clearly outline the main biomarkers identified in this study and explain how KIFC3, in combination with other pathways, plays a role in CRC. Thank you again for your thorough review.

i don't see future direction mentioned in the paper

Re: Thank you for reviewing our study and providing valuable feedback. Based on your suggestions, we have added a section at the end of the discussion (lines 500-505) outlining plans for further validation of the proposed mechanisms. We intend to conduct cell and animal experiments to further investigate the molecular mechanisms related to the identified biomarkers, including pathways or immune cells. Once again, thank you for your invaluable input, which is crucial to our research.

the introduction does not clarify how this research novelty differs from previous work on KIFs in cancer.

Re: Thank you for reviewing our study and providing invaluable feedback. Based on your suggestions, we have added a discussion in the introduction about the primary mechanisms by which KIFs function in CRC (lines 76-78). However, the specific molecular mechanisms through which KIF members improve CRC patient prognosis still require further investigation. Once again, thank you for your valuable input, which is essential to our research.

Reviewer 2 Report

Comments and Suggestions for Authors

Title: Single-cell transcriptome joint exploration of biomarkers re-lated to driver protein families in colorectal cancer and mecha-nistic studies.

Comments:

  1. Which gene of KIFC2 and KIFC3 shows high survival rate and why? Include these details in the manuscript.
  2. Are the mentioned drugs in the abstract are KIFC2/KIFC3 inhibitors? Also is the mentioned drugs have any role in NK T cells?
  3. Authors should plan to include GEPIA2 bioinformatic database. GEPIA2 database provide the expression of the target gene in normal and tumor tissue.
  4. Include the future perspective in the last section.

Author Response

Comments:

1.Which gene of KIFC2 and KIFC3 shows high survival rate and why? Include these details in the manuscript.

Re: Thank you for reviewing our study and providing valuable feedback. Based on our results, we found that KIFC3 is highly associated with CRC patient survival. In response to your comments, we have revised the description of the results in section 3.2 (lines 278-282) and modified the discussion regarding the relationship between KIFC3 and CRC patient survival (lines 410-412). Thank you again for your invaluable input.

2.Are the mentioned drugs in the abstract are KIFC2/KIFC3 inhibitors? Also is the mentioned drugs have any role in NK T cells?

Re:  Thank you for reviewing our study and providing valuable feedback. Based on your comments, we recognized that our previous wording in the abstract might have been misleading. We have now modified the phrase “in the treatment of CRC” to “in the drug sensitivity of CRC” to clarify that this analysis specifically pertains to drug sensitivity. Additionally, in the discussion (line 457-461), we have clarified the effects of drugs on NK and T cells, and we speculate that these drugs may exert their therapeutic effects by influencing certain functions of NK or T cells. Once again, thank you for your invaluable input.

3.Authors should plan to include GEPIA2 bioinformatic database. GEPIA2 database provide the expression of the target gene in normal and tumor tissue.

Re: Thank you very much for reviewing our study and providing valuable feedback. Based on your suggestions, we have updated the Materials and Methods (section 2.5, lines 132-134) and Results (3.2, lines 282-289) sections to include the expression levels and survival differences of KIFC2 and KIFC3 in the GEPIA2 database. The results show that in the GEPIA2 database, KIFC2 expression is significantly lower in rectal cancer (READ) compared to the control group, while KIFC3 expression is significantly higher in colon cancer (COAD) compared to the control group. Additionally, patients with low KIFC3 expression had a significantly higher survival rate than those with high KIFC3 expression, whereas no significant survival difference was observed between the high and low KIFC2 expression groups. Once again, we sincerely appreciate your thorough review and look forward to receiving your further feedback.

4.Include the future perspective in the last section.

Re: Thank you for reviewing our research and providing your invaluable suggestions. Based on your feedback, we have added additional validation of the underlying mechanisms related to this study in the final section of the discussion (lines 500–505). We plan to further verify the molecular mechanisms—whether via signaling pathways or immune cell interactions—associated with the biomarkers using cellular or animal experiments. Once again, we truly appreciate your insightful comments, which are critical to our research.

Reviewer 3 Report

Comments and Suggestions for Authors

The title of this manuscript is not properly reflecting its content; please revise and change to a more specific one.

The aim of the work is not clear; please clarify in the abstract and the final part of the introduction.

In the paper methodology, write clearly which parts have been done in silico and which have been performed in vitro.

In general, the abbreviation system needs to be fixed for better perception by readers. Full name should be written on first mention within the text, at the beginning of each paragraph, and within the figure legends.

Regarding KIF, authors should specify in every occasion. When referring to the family, it should be written as “KIF members,” and certain members should be listed by their definite names.

Minor remarks

Abstract: Background: Studies suggest that kinesin family members (KIF)... change to “kinesin family (KIF) members”

Introduction:

Moreover, conventional CRC diagnostic markers such as CA19-9 exhibit limited sensitivity for early detection[3]. Write CA19-9 in full.

This study aimed to identify and analyze CRC biomarkers of CRC, providing a reference for the treatment and prognosis management of CRC. Move this sentence to the end of the introduction.

The kinesin family (KIF), a pivotal intracellular transport molecular motor, plays crucial roles in mitosis, material conveyance, and other physiological processes[4]. Change to “The kinesin family (KIF) of proteins, ...”

By leveraging CRC-related data from The Cancer Genome Atlas (TCGA) and Gene

Expression Omnibus (GEO) databases, this study employed bioinformatics methodologies to identify KIFC3 as a key biomarker associated with KIF, thus highlighting its potential for predicting CRC progression and aiding the discovery of effective therapeutic

targets for the disease. Through functional enrichment analysis, immune infiltration assessment, regulatory network elucidation, and drug sensitivity profiling of these markers, insights into the potential connections and regulatory pathways between CRC and KIF were obtained. Notably, the integration of single-cell analysis technology has facilitated profound exploration of CRC mechanisms at the cellular level, offering robust theoretical underpinnings for early CRC diagnosis, identification of therapeutic targets, and personalized medical interventions. The findings of this study hold promise for introducing novel approaches to clinical CRC management, thereby providing patients with more effective and tailored medical care. THIS PARAGRAPH IS CONFUSING FOR READERS; authors may trim unnecessary information and focus mainly on study aims and give a very brief description of the methodology used.

Material and methods

2.3. The function analysis of intersection genes change to “Functional analysis of the intersection genes”

2.4. The survival analysis of intersection genes, Delete “The”

2.13. Descriptions about HE staining are not necessarily to be written here; just write the tumor identity was identified in HE-stained section prior to processing tumor samples for DNA extraction.

Results

3.1. write DEGs, DEMGs, and KIFRGs in full

Figure 1 legend: the font of panel B, C, and E is small for the human eye.

Figure 2 legend: write “GO” and “KEEG” in full. Remove the small letter “c” on the right panel of Figure 2B

Figure 3 legend: the font of all figure panels is too small for the human eye.

Figure 4 legend: write KIFC3 in full.

3.4. The constructed regulatory networks provided valuable insights into the potential mechanisms for treating CRC, revise and trim this title. Also write TFs in full in the text.

Figure 7 legend: write t-SNE in full.

3.10. Expression of KIF3C in CRC tissues and normal tissues

KIF3C expression was observed to be low in tissues from 25 CRC cases, whereas it was significantly higher in the corresponding normal tissues, indicating a statistically significant difference (Figure 10). Rewrite this part.

Figure 10. QPCR analysis of biomarker expression in CRC and normal tissue. Replace the word “biomarker” with “KIF3C”

5. Conclusion: this part it overly written and too optimistic. Since the decrease of KIFC3 is modest in colorectal cancer samples, as shown by the qPCR analysis of the present study, retrieving its full function might be not solely enough to treat the disease but rather to work in conjunction with other factors.

Comments on the Quality of English Language

The English of the manuscript needs carefule revision for the abbreviations and syntax.

Author Response

The title of this manuscript is not properly reflecting its content; please revise and change to a more specific one.

Re: Thank you for reviewing our study and providing your invaluable suggestions. Based on your feedback, we have revised the title of our research to “Identification of KIFC3 as a Colorectal Cancer Biomarker and Its Regulatory Mechanism in the Immune Microenvironment Based on Integrated Analysis of Multi-Omics Databases” (lines 2–4). Once again, we truly appreciate your valuable input, which is critical to our work.

The aim of the work is not clear; please clarify in the abstract and the final part of the introduction.

Re: Thank you for reviewing our study and providing your invaluable suggestions. Based on your feedback, we have revised the research objective in the background section of the abstract (lines 20–21). Once again, we truly appreciate your valuable input, which is essential to our work.

In the paper methodology, write clearly which parts have been done in silico and which have been performed in vitro.

Re: Thank you for reviewing our study and providing your invaluable suggestions. Based on your feedback, we have reexamined our analytical and statistical methods and have clearly specified the datasets and experiments in the methods section headings. Once again, we sincerely appreciate your valuable input.

In general, the abbreviation system needs to be fixed for better perception by readers. Full name should be written on first mention within the text, at the beginning of each paragraph, and within the figure legends.

Re: Thank you for reviewing our study and providing your invaluable suggestions. Based on your feedback, we have reviewed and revised the abbreviations and acronyms used throughout the text. Once again, we sincerely appreciate your valuable input.

Regarding KIF, authors should specify in every occasion. When referring to the family, it should be written as “KIF members,” and certain members should be listed by their definite names.

Re: Thank you for reviewing our study and providing your invaluable suggestions. Based on your feedback, we have replaced "KIF" with "KIF members" throughout the manuscript. Once again, we sincerely appreciate your valuable input.

Minor remarks

Abstract: Background: Studies suggest that kinesin family members (KIF)... change to “kinesin family (KIF) members”

Re: Thank you for reviewing our study and providing your invaluable suggestions. Based on your feedback, we have revised “kinesin family members (KIF)” to “kinesin family (KIF) members” throughout the manuscript (lines 18, 62). Once again, we truly appreciate your valuable input.

Introduction:

Moreover, conventional CRC diagnostic markers such as CA19-9 exhibit limited sensitivity for early detection[3]. Write CA19-9 in full.

Re: Thank you for reviewing our study and providing your invaluable suggestions. Based on your feedback, we have revised "CA19-9" to "Carbohydrate Antigen 19-9 (CA19-9)" in the manuscript (lines 57). Once again, we truly appreciate your valuable input.

This study aimed to identify and analyze CRC biomarkers of CRC, providing a reference for the treatment and prognosis management of CRC. Move this sentence to the end of the introduction.
Re: Thank you for reviewing our study and providing your invaluable suggestions. Based on your feedback, we have revised the concluding part of the introduction (lines 84–85). Once again, we truly appreciate your valuable input.

The kinesin family (KIF), a pivotal intracellular transport molecular motor, plays crucial roles in mitosis, material conveyance, and other physiological processes[4]. Change to “The kinesin family (KIF) of proteins, ...”

Re: Thank you for reviewing our study and providing your invaluable suggestions. Based on your feedback, we have revised the introduction from “The kinesin family (KIF)” to “The kinesin family (KIF) of proteins” (line 62). Once again, we sincerely appreciate your valuable input.

By leveraging CRC-related data from The Cancer Genome Atlas (TCGA) and Gene Expression Omnibus (GEO) databases, this study employed bioinformatics methodologies to identify KIFC3 as a key biomarker associated with KIF, thus highlighting its potential for predicting CRC progression and aiding the discovery of effective therapeutic targets for the disease. Through functional enrichment analysis, immune infiltration assessment, regulatory network elucidation, and drug sensitivity profiling of these markers, insights into the potential connections and regulatory pathways between CRC and KIF were obtained. Notably, the integration of single-cell analysis technology has facilitated profound exploration of CRC mechanisms at the cellular level, offering robust theoretical underpinnings for early CRC diagnosis, identification of therapeutic targets, and personalized medical interventions. The findings of this study hold promise for introducing novel approaches to clinical CRC management, thereby providing patients with more effective and tailored medical care. THIS PARAGRAPH IS CONFUSING FOR READERS; authors may trim unnecessary information and focus mainly on study aims and give a very brief description of the methodology used.

Re: Thank you for reviewing our study and providing your invaluable suggestions. Based on your feedback, we have simplified the final paragraph of the introduction (lines 80–89). Once again, we sincerely appreciate your valuable input.

Material and methods

2.3. The function analysis of intersection genes change to “Functional analysis of the intersection genes”

Re: Thank you for reviewing our study and providing your invaluable suggestions. Based on your feedback, we have revised the title of the methods section in section 2.3. Once again, we sincerely appreciate your valuable input.

2.4. The survival analysis of intersection genes, Delete “The”

Re: Thank you for reviewing our study and providing your invaluable suggestions. Based on your feedback, we have revised the title of the methods section in Section 2.4 (line 124). Once again, we sincerely appreciate your valuable input.

2.13. Descriptions about HE staining are not necessarily to be written here; just write the tumor identity was identified in HE-stained section prior to processing tumor samples for DNA extraction.

Re: Thank you for reviewing our study and providing your invaluable suggestions. Based on your feedback, we have removed the excessive HE description from the methods section in Section 2.13. Once again, we sincerely appreciate your valuable input.

Results

3.1. write DEGs, DEMGs, and KIFRGs in full

Figure 1 legend: the font of panel B, C, and E is small for the human eye.

Figure 2 legend: write “GO” and “KEEG” in full. Remove the small letter “c” on the right panel of Figure 2B

Figure 3 legend: the font of all figure panels is too small for the human eye.

Figure 4 legend: write KIFC3 in full.

Re: Thank you for reviewing our study and providing your invaluable suggestions. Based on your feedback, we have re-examined and revised the terms “DEGs,” “DEMGs,” and “KIFRGs” in the text (line 260-261, 271-274, 303), and have also modified the text in the figures. Once again, we sincerely appreciate your valuable input.

3.4. The constructed regulatory networks provided valuable insights into the potential mechanisms for treating CRC, revise and trim this title. Also write TFs in full in the text.

Re: Thank you for reviewing our study and providing your invaluable suggestions. Based on your feedback, we have revised the title of section 3.4 (line 310) and verified the full name of TFs. Once again, we sincerely appreciate your valuable input.

Figure 7 legend: write t-SNE in full.

Re: Thank you for reviewing our study and providing your invaluable suggestions. Based on your feedback, we have included the full name of t-SNE in the legend of Figure 7 (line 347). Once again, we sincerely appreciate your valuable input.

3.10. Expression of KIF3C in CRC tissues and normal tissues

KIF3C expression was observed to be low in tissues from 25 CRC cases, whereas it was significantly higher in the corresponding normal tissues, indicating a statistically significant difference (Figure 10). Rewrite this part.

Re: Thank you for reviewing our study and providing your invaluable suggestions. Based on your feedback, we have revised the results section in Section 3.10 (lines 389–390). Once again, we sincerely appreciate your valuable input.

Figure 10. QPCR analysis of biomarker expression in CRC and normal tissue. Replace the word “biomarker” with “KIF3C”

Re: Thank you for reviewing our study and providing your invaluable suggestions. Based on your feedback, we have revised the results section in Section 3.10 by changing “biomarker” to “KIFC3” in the figure legend (line 392). Once again, we sincerely appreciate your valuable input.

Conclusion: this part it overly written and too optimistic. Since the decrease of KIFC3 is modest in colorectal cancer samples, as shown by the qPCR analysis of the present study, retrieving its full function might be not solely enough to treat the disease but rather to work in conjunction with other factors.

Re: Thank you for reviewing our study and providing your invaluable suggestions. Based on your feedback, we have recognized the limitations of our research and revised the conclusions section (lines 507–513) to emphasize that KIFC3 may exert its effects in CRC by influencing or modulating other pathways (signaling or immune cells), thereby offering insights for clinical treatment of CRC patients. Once again, we sincerely appreciate your valuable input.

Comments on the Quality of English Language

The English of the manuscript needs carefule revision for the abbreviations and syntax.

Re: Thank you for reviewing our study and providing your invaluable suggestions. Based on your feedback, we have refined the entire manuscript. Once again, we sincerely appreciate your valuable input, which is crucial to our paper.

Reviewer 4 Report

Comments and Suggestions for Authors

I have reviewed the manuscript titled "Single-Cell Transcriptome Joint Exploration of Biomarkers Related to Driver Protein Families in Colorectal Cancer and Mechanistic Studies" . The study explores an important topic by investigating the role of Kinesin family genes in colorectal cancer (CRC) using single-cell transcriptomics and bioinformatics approaches. While the research provides valuable insights, the manuscript requires revisions before it can be considered for publication. Some areas need improvement, including  justification of experimental design and stronger emphasis on clinical implications. Below, I provide detailed comments and suggestions to enhance the clarity, structure, and scientific rigor of the manuscript.

  • Line 6-10: The objective can be reworded for clarity: e.g:
    “This study investigates the role of Kinesin family genes in colorectal cancer (CRC) using single-cell transcriptomics and bioinformatics approaches.”
  • Line 17-20: The description of methods should be simplified—avoid excessive detail on database sources.
  • Line 25-30: The conclusion should explicitly state why KIFC3 is a promising biomarker and how it could impact CRC diagnosis or treatment.
  • Line 42-50: The global incidence of CRC should be summarized in a single sentence.
  • Line 56-68: The role of KIF family proteins should be linked more explicitly to CRC rather than listing general functions.
  • Line 79-85: Clearly state the hypothesis, e.g.:
    “We hypothesize that KIFC3 functions as a key oncogene in CRC and may serve as a potential biomarker for diagnosis and treatment.”
  • Line 86-90: Provide a stronger justification for using single-cell RNA sequencing (scRNA-seq) over bulk RNA-seq.
  • Line 95-110: Explain why KIFC3 was selected over other candidate genes.
  • Line 185-190: Clarify how KIFC3 expression correlates with patient survival in multiple datasets.
  • Line 230-240: The discussion on immune infiltration should explain how KIFC3 might influence the tumor microenvironment.
  • Line 275-280: Suggest how KIFC3 could be integrated into future CRC diagnostic tests or targeted therapies.

Author Response

I have reviewed the manuscript titled "Single-Cell Transcriptome Joint Exploration of Biomarkers Related to Driver Protein Families in Colorectal Cancer and Mechanistic Studies" . The study explores an important topic by investigating the role of Kinesin family genes in colorectal cancer (CRC) using single-cell transcriptomics and bioinformatics approaches. While the research provides valuable insights, the manuscript requires revisions before it can be considered for publication. Some areas need improvement, including  justification of experimental design and stronger emphasis on clinical implications. Below, I provide detailed comments and suggestions to enhance the clarity, structure, and scientific rigor of the manuscript.

Line 6-10: The objective can be reworded for clarity: e.g:
“This study investigates the role of Kinesin family genes in colorectal cancer (CRC) using single-cell transcriptomics and bioinformatics approaches.”

Re: Thank you for reviewing our study and providing your invaluable suggestions. Based on your feedback, we have revised the objective section of the abstract (lines 20–21). Once again, we sincerely appreciate your valuable input.

The description of methods should be simplified—avoid excessive detail on database sources.

Re: Thank you for reviewing our study and providing your invaluable suggestions. Based on your feedback, we have revised the methods section of the abstract (lines 26–28). Once again, we sincerely appreciate your valuable input.

The conclusion should explicitly state why KIFC3 is a promising biomarker and how it could impact CRC diagnosis or treatment.

Re: Thank you for reviewing our study and providing your invaluable suggestions. Based on your feedback, we have clarified in the abstract the rationale for using KIFC3 as a biomarker, explaining that it influences the prognosis of CRC patients through the tumor microenvironment (lines 42–44). Once again, we sincerely appreciate your valuable input.

The global incidence of CRC should be summarized in a single sentence.

Re: Thank you for reviewing our study and providing your invaluable suggestions. Based on your feedback, we have revised the description of CRC in the introduction (lines 51–53) to eliminate redundant information. Once again, we sincerely appreciate your valuable input.

The role of KIF family proteins should be linked more explicitly to CRC rather than listing general functions.

Re: Thank you for reviewing our study and providing your invaluable suggestions. Based on your feedback, we have expanded on the role of KIF family proteins in CRC within the introduction (lines 67–72), emphasizing the mechanisms by which KIF influences CRC. Once again, we sincerely appreciate your valuable input.

Clearly state the hypothesis, e.g.:
“We hypothesize that KIFC3 functions as a key oncogene in CRC and may serve as a potential biomarker for diagnosis and treatment.”

Re: Thank you for reviewing our study and providing your invaluable suggestions. Based on your feedback, we have revised the conclusion section of the introduction (lines 87–91) to emphasize the study's hypothesis. Once again, we sincerely appreciate your valuable input.

Provide a stronger justification for using single-cell RNA sequencing (scRNA-seq) over bulk RNA-seq.

Re: We sincerely thank the editor and all reviewers for their valuable feedback that. Based on your feedback, we have added a section to the introduction highlighting the advantages of single-cell analysis (lines 84–86). We now explain that single-cell sequencing technology enables us to analyze gene expression at the individual cell level, which is crucial for identifying subtle differences in cell subpopulations and cell states. Once again, we sincerely appreciate your valuable input.

Explain why KIFC3 was selected over other candidate genes.

Re: Thank you for reviewing our study and providing your invaluable suggestions. KIFC3, the primary biomarker in our research, was identified based on CRC datasets and KIF family-related genes through differential gene expression, differential methylation, and survival analyses. Notably, KIFC3 exhibited significant differences between CRC and control samples and demonstrated a strong association with CRC patient survival, as evidenced by the marked survival differences between the high and low KIFC3 expression groups. Once again, we sincerely appreciate your valuable input.

Clarify how KIFC3 expression correlates with patient survival in multiple datasets.

Re: Thank you for pointing this out. By comparing survival differences between the high and low KIFC3 expression groups across multiple databases (notably, the GSE39582 dataset and the GEPIA2 database demonstrated significant survival differences between the high and low KIFC3 expression groups, while no significant survival differences were observed for KIFC2), these results further confirm the strong association between KIFC3 expression and CRC patient survival (lines 280–292). Once again, we sincerely appreciate your valuable input.

The discussion on immune infiltration should explain how KIFC3 might influence the tumor microenvironment.

Re:  We feel great thanks for your professional review work on our article. Based on your feedback, we have expanded the discussion on immune infiltration to include the role of KIFC3 in association with follicular helper T cells in osteosarcoma metastasis. Combined with our findings, we further hypothesize an immunotherapeutic role for KIFC3 in CRC (line 441-446). Once again, we sincerely appreciate your valuable input.

Suggest how KIFC3 could be integrated into future CRC diagnostic tests or targeted therapies.

Re: Thank you for reviewing our study and providing your invaluable suggestions. Based on your feedback, we have revised the conclusions section regarding the interaction between KIFC3 and immune cells in the tumor microenvironment of CRC patients, demonstrating its potential as an immunotherapeutic target for precision treatment of CRC (lines 512–515). Clinically, KIFC3 could be targeted by developing CAR-T (chimeric antigen receptor T cell) or CAR-NK (chimeric antigen receptor NK cell) therapies. A CAR construct that recognizes KIFC3 would be engineered and transduced into T cells or NK cells, which are then expanded in vitro and reinfused into the patient to selectively eliminate tumor cells expressing KIFC3. Furthermore, combining this approach with other immunomodulatory factors or cytokines could enhance the overall efficacy of the treatment. Once again, we sincerely appreciate your valuable input.

Round 2

Reviewer 1 Report

Comments and Suggestions for Authors

Thank you for answering my questions. Good luck